# Cinnamaldehyde-Treated Bone Marrow Mesenchymal-Stem-Cell-Derived Exosomes via Aqueous Two-Phase System Attenuate IL-1β-Induced Inflammation and Catabolism via Modulation of Proinflammatory Signaling Pathways

**DOI:** 10.3390/ijms25137263

**Published:** 2024-07-01

**Authors:** Jaishree Sankaranarayanan, Seok Cheol Lee, Hyung Keun Kim, Ju Yeon Kang, Sree Samanvitha Kuppa, Jong Keun Seon

**Affiliations:** 1Department of Biomedical Sciences, Chonnam National University Medical School, Hwasun 58128, Republic of Korea; jaanu.p2206@gmail.com (J.S.); sreesamanvitha95@gmail.com (S.S.K.); 2Department of Orthopaedic Surgery, Center for Joint Disease of Chonnam National University Hwasun Hospital, 322 Seoyang-ro, Hwasun-eup 519763, Republic of Korea; iwannaseeu@naver.com (S.C.L.); chemokines@naver.com (H.K.K.); jy0194@naver.com (J.Y.K.); 3Korea Biomedical Materials and Devices Innovation Research Center, Chonnam National University Hospital, 42 Jebong-ro, Dong-gu, Gwangju 501757, Republic of Korea

**Keywords:** exosomes, osteoarthritis, chondrocytes, aqueous two-phase systems, cinnamaldehyde, anti-inflammation

## Abstract

Osteoarthritis (OA) is a degenerative joint disorder that is distinguished by inflammation and chronic cartilage damage. Interleukin-1β (IL-1β) is a proinflammatory cytokine that plays an important role in the catabolic processes that underlie the pathogenesis of OA. In this study, we investigate the therapeutic efficacy of exosomes derived from untreated bone-marrow-derived mesenchymal stem cells (BMMSC-Exo) and those treated with cinnamaldehyde (BMMSC-CA-Exo) for preventing the in vitro catabolic effects of IL-1β on chondrocytes. We stimulated chondrocytes with IL-1β to mimic the inflammatory microenvironment of OA. We then treated these chondrocytes with BMMSC-Exo and BMMSC-CA-Exo isolated via an aqueous two-phase system and evaluated their effects on the key cellular processes using molecular techniques. Our findings revealed that treatment with BMMSC-Exo reduces the catabolic effects of IL-1β on chondrocytes and alleviates inflammation. However, further studies directly comparing treatments with BMMSC-Exo and BMMSC-CA-Exo are needed to determine if CA preconditioning can provide additional anti-inflammatory benefits to the exosomes beyond those of CA preconditioning or treatment with regular BMMSC-Exo. Through a comprehensive molecular analysis, we elucidated the regulatory mechanisms underlying this protective effect. We found a significant downregulation of proinflammatory signaling pathways in exosome-infected chondrocytes, suggesting the potential modulation of the NF-κB and MAPK signaling cascades. Furthermore, our study identified the molecular cargo of BMMSC-Exo and BMMSC-CA-Exo, determining the key molecules, such as anti-inflammatory cytokines and cartilage-associated factors, that may contribute to their acquisition of chondroprotective properties. In summary, BMMSC-Exo and BMMSC-CA-Exo exhibit the potential as therapeutic agents for OA by antagonizing the in vitro catabolic effects of IL-1β on chondrocytes. The regulation of the proinflammatory signaling pathways and bioactive molecules delivered by the exosomes suggests a multifaceted mechanism of action. These findings highlight the need for further investigation into exosome-based therapies for OA and joint-related diseases.

## 1. Introduction

Chondrocytes, which are essential cellular components of articular cartilage, are crucial for maintaining cartilage homeostasis and ensuring its structural integrity. These specialized cells synthesize and secrete extracellular matrix (ECM) components such as collagen and proteoglycans, thereby providing mechanical support and shock absorption to joints. In addition, chondrocytes regulate cartilage metabolism by maintaining the equilibrium between anabolic and catabolic processes [1]. However, factors such as inflammation can disrupt their normal functioning and contribute to cartilage degradation. The in vitro inflammation of chondrocytes can be elicited by treatment with proinflammatory cytokines such as tumor necrosis factor-alpha (TNF-α) and interleukin-1β (IL-1β), which leads to the production of matrix metalloproteinases (MMPs) and aggrecanases that degrade ECM components. Due to the detrimental effects of inflammation on chondrocytes and cartilage health, researchers are increasingly exploring new potential therapeutic strategies to mitigate inflammation in these cells, which have been implicated in several cartilage-related disorders, including osteoarthritis (OA) [2]. An array of multifactorial etiologies including distress, sexuality, age, endocrine levels, obesity, and congenital traits can induce the onset of the degenerative joint disease of OA. This disease is characterized by cartilage deterioration or fissure formation, matrix oscillation, inflammation of the synovium, osteophyte origination, and subchondral bone sclerosis [3].

OA is a generic form of arthritis that affects millions of people worldwide. In 2019, approximately 528 million people lived with OA, which is a 113.25% increase since 1990, and there are clear age, sex, geographic diversity, and anatomic site disparities [4]. According to recent studies, the global prevalence of OA is 7% [5]. Dysregulated chondrocyte hypertrophy caused by proinflammatory cytokines in OA contributes to the pathological remodeling of the ECM, cartilage degradation, and joints. In addition, various immune factors are associated with the pathogenesis of OA, and extensive research indicates that IL-1β is a key player in the development and progression of OA. Its potent proinflammatory effects and ability to induce catabolic processes in chondrocytes make IL-1β an important immune factor in the pathogenesis of OA. Elevated levels of IL-1β are found in the cartilage, subchondral bone, joint fluid, and synovial membranes of patients with OA. Patients with OA exhibit a higher expression of IL-1β in their chondrocytes than healthy individuals; this accelerates disease progression through the upregulation of catabolic factors [6]. Although there is no cure available yet for OA, there are several treatments that can help people with OA manage their symptoms and enhance their quality of life. Exercise, weight loss, physical therapy, and assistive technology, such as canes or braces, are examples of nonpharmacologic treatment approaches. Painkillers, such as nonsteroidal anti-inflammatory drugs (NSAIDs), topical gels and creams, and intra-articular injections of hyaluronic acid or corticosteroids, are used as pharmacologic therapies [7]. Joint fusion, osteotomy, and joint replacement surgery are among the surgical alternatives [8]. The projected global prevalence of OA in 2050 is predicted to be significantly higher than the current prevalence, thus emphasizing the need for effective prevention and treatment strategies [5].

Exosomes have emerged as promising candidates for the treatment of inflammatory conditions such as OA [9]. These small extracellular vesicles are secreted by various cell types, including chondrocytes, and participate in intercellular communication by transferring bioactive molecules to recipient cells. Their unique properties make them attractive therapeutic agents for modulating inflammation in chondrocytes [10]. Treatment strategies highlight a growing interest in exploring the use of natural compounds as potential therapeutic options for the management of OA. Compounds derived from plants and natural sources, such as cinnamaldehyde (CA) from cinnamon, *Boswellia serrata*, *Arnica montana*, *honeybee products*, *Psoralea corylifolia*, and *Rhizoma coptidis*, have exhibited promising anti-inflammatory and chondroprotective effects in various OA models [11]. These natural compounds have been shown to modulate key signaling pathways involved in joint inflammation and cartilage degradation including the NF-κB, PI3K/Akt/mTOR, and MMP pathways [12]. In addition, exosomes derived from cells treated with natural compounds, such as CA, may inherit the beneficial properties of these compounds, thereby offering a novel approach for delivering targeted therapy to osteoarthritic joints. While the use of natural compounds holds promise, careful consideration must be given to the potential side effects, such as hepatotoxicity, nephrotoxicity, drug interactions, and long-term safety concerns. Ongoing research aims to elucidate their mechanisms of action further and optimize the therapeutic potential of these natural compounds for the management of OA. Several studies have reported the anti-inflammatory effects of exosomes derived from different cell sources in various disease models [13]. However, the specific impact of exosomes on inflammation in chondrocytes is currently being actively investigated. Methods to isolate exosomes include ultracentrifugation, size-exclusion chromatography, immunoaffinity capture, microfluidics, and polymeric precipitation. Ultracentrifugation has limitations because of its time-consuming nature and low yield. Size-exclusion chromatography often results in a loss of exosomes because they adhere to the filters. Immunoaffinity isolation techniques use antibodies but are impractical because of their expensive physical contact requirements [14]. Finally, microfluidic methods have been explored using immunoaffinity or size exclusion but have low efficiency and are inappropriate for pre-analytical processes [15].

Exosomes are extracellular vesicles that have been shown to play a vital role in intercellular communication and have potential as biomarkers [16]. However, the isolation and purification of exosomes remain challenging because of low yields and poor-efficiency techniques. In this regard, an aqueous two-phase system (ATPS) has been suggested as a promising technique for exosome isolation and application in therapeutics [17]. Two incompatible polymers, such as polyethylene glycol (PEG) and dextran (DEX), are dispersed to form an ATPS. The physicochemical association between the molecules in the sample mixture, the phase-forming chemicals, and other system design factors provide an environment in which the molecules separate to either the top or bottom phase when a specific combination is added to the process [18]. An ATPS has many advantages over conventional separation methods, such as being environmentally friendly, low-cost, sustainable, and easy to use [19]. In recent studies, an ATPS has been used to extract highly pure ECM components including exosomes from a mixture of extracellular vesicles (EVs) and serum proteins. PEG/DEX ATPS has been used to isolate EVs [20], and the amount of PEG variation between the two-phase and monophasic systems decreased as the DEX ratio increased. An ATPS has also been used to encapsulate cells and to isolate and purify IgG. The purification of target molecules can be accomplished under the right conditions, whereby they move to a different phase from the contaminants [21]. Since phosphate-buffered saline (PBS) makes up the majority of an ATPS, this purification method offers intriguing benefits for the recovery and purification of exosomes. Overall, an ATPS is a promising tool for exosome isolation and application in therapeutics. Although diverse methods for isolating and purifying exosomes have been established, they have some shortfalls and cannot address all the requirements. The separation impact of several combined isolation techniques is probably superior to that of a single method. Therefore, numerous research teams have begun combining different techniques to separate and purify exosomes to enhance yield and purity and improve the separation efficiency and enrichment to acquire the optimal exosomes. CA has been shown to dramatically suppress lipopolysaccharide (LPS)-stimulated NF-κB activation in chondrocytes [22], and trans-cinnamaldehyde (TCA) has been shown to inhibit IL-1β-induced inflammation in chondrocytes. It also decreases the mRNA levels of iNOS, MMP-13, ADAMTS-5, and COX-2, which are enhanced by IL-1β treatment [23]. As the potent anti-inflammatory properties of CA have been demonstrated in recent studies, it is reasonable to hypothesize that encapsulating CA in exosomes could further enhance its therapeutic potential for OA by improving the delivery and bioavailability to chondrocytes.

This study aims to investigate the anti-inflammatory properties of exosomes derived from bone-marrow-derived mesenchymal stem cells (BMMSC-Exo) and those treated with CA (BMMSC-CA-Exo) and to evaluate their impact on the expression of proinflammatory cytokines and matrix-degrading enzymes in chondrocytes. In particular, the study aims to evaluate the effects of exosomes isolated via an ATPS on decreasing inflammation in chondrocytes. Furthermore, the effects of exosomes isolated with and without CA treatment from BMMSCs on OA were assessed using an IL-1β-induced OA cell model using chondrocytes. PCR, immunofluorescence, and western-blotting techniques were used to assess the inflammatory panel of the gene expression, protein expression, and phosphorylation levels of ERK, JNK, and MAPK, and NF-κB signal activation in chondrocytes that were stimulated with IL-1β and treated with BMMSC-Exo. The inhibition of p38 MAPK and NF-κB signal activation in response to IL-1β stimulation in chondrocytes treated with BMMSC-Exo and BMMSC-CA-Exo was also confirmed. The findings from this study contribute to our understanding of the exosome-mediated regulation of inflammation in chondrocytes and provide a basis for the development of novel therapeutic approaches for acute cartilage-related disorders and chronic OA.

## 2. Results

### 2.1. BMMSC Phenotype and Exosome Characterization

To investigate the effect of BMMSC-Exo and BMMSC-CA-Exo on chondrocytes, exosomes derived from BMMSCs were isolated and identified by a nanoparticle tracking analysis (NTA), transmission electron microscopy (TEM), and western blot. According to the NTA results of the exosome diameter distribution, the particle diameters varied from 30 to 150 nm (Figure 1E,F). TEM was used to analyze the size and structure of isolated exosomes (Figure 1G). The western blot analysis was then used to identify the expression of the ALIX, CD63, and CD81 exosome markers in the separated exosomes (Figure 1H). Based on these results, it was concluded that exosomes were successfully isolated and purified from the BMMSCs.

### 2.2. Effect of BMMSC-Exo and BMMSC-CA-Exo on Cell Viability

A CCK-8 assay was used to evaluate the effects of exosomes on chondrocyte cell viability under inflammatory conditions. The results showed that exosomes at concentrations below 5 mg/mL did not significantly affect cell viability compared with the control group (Figure 2A,B) Thus, exosomes may not affect chondrocyte cell viability in in vitro inflammatory environments. (Refer Appendix A for Live-Dead analysis)

### 2.3. Promotion of Chondrocyte Migration by BMMSC-Exo and BMMSC-CA-Exo Treatment

As shown in Figure 3, treatment with 10 ng/mL of IL-1β significantly inhibited chondrocyte migration. However, this effect could be reversed by co-culturing the chondrocytes with BMMSC-Exo and BMMSC-CA-Exo. Moreover, chondrocytes treated with BMMSC-Exo exhibited a higher number of migratory cells than the group treated with IL-1β. The scratch wound assays were used to detect chondrocyte motility, and the results indicated that, after being induced with 10 ng/mL of IL-1β, only the treatment with 5 μg/mL of BMMSC-Exo and BMMSC-CA-Exo significantly enhanced chondrocyte motility at 0, 24, and 48 h. Cell migration was evaluated at 0, 24, and 48 h, as the 12 h timepoint did not show significant migration.

### 2.4. Effects of BMMSC-Exo and BMMSC-CA-Exo on IL-1β-Induced ECM Degradation

We measured the mRNA expression levels of COL-II, a marker unique to cartilage in chondrocytes. COL-II is a major component of the ECM and plays a key role in cartilage stability. Matrix-degrading enzymes, such as MMPs, trigger ECM breakdown during OA. To assess chondrocyte deterioration, we used the western blot and immunofluorescence analysis to determine the effect of BMMSC-Exo and BMMSC-CA-Exo treatment on the COL-II levels in IL-1β-treated chondrocytes. COL-II levels were drastically decreased following IL-1β treatment, but this was restored by a post-treatment with 5 μg/mL of BMMSC-Exo and BMMSC-CA-Exo (Figure 4A,B). Furthermore, immunofluorescence examination revealed that, unlike the IL-1β group, treatment with BMMSC-Exo and BMMSC-CA-Exo prevented IL-1β-stimulated cytoplasmic COL-II degradation (Figure 4E).

### 2.5. Effect of BMMSC-Exo and BMMSC-CA-Exo Treatment on Inflammation Markers and NF-κB Signal Activation

The expression of the inflammatory factors IL-6, MMP-13, COX-2, and NF-κB was measured by a western blot analysis and PCR (Figure 4C,D). We investigated whether the protective effects of BMMSC-Exo and BMMSC-CA-Exo treatment on chondrocytes were related to the modulation of the NF-κB signaling pathway in cells treated with IL-1β. To examine this, we used western blotting to analyze the phosphorylated and total protein levels of p65 in the whole-cell lysates. We used 5 g/mL of BMMSC-Exo and BMMSC-CA-Exo, which were revealed to have the greatest effect in decreasing hypertrophic and inflammatory mediators. When compared with the control group, IL-1β treatment significantly increased the expression of the NF-κB subunit p65 (pp65), indicating that the pp65 signaling pathway was activated (Figure 5A,B). However, exosome treatment of the IL-1β-exposed chondrocytes effectively suppressed the induced expression of pp65, and treatment with BMMSC-Exo and BMMSC-CA-Exo prevented the release of inflammatory mediators and cytokines by blocking the activation of the NF-κB pathway in chondrocytes, suggesting that exosomes tend to inhibit the NF-κB signaling pathway. To further investigate the mechanism of action, we employed 5HPP-33, a specific inhibitor of the NF-κB pathway. According to our research, 5HPP-33 halted pp65 from being triggered by IL-1β and partially restricted the activation of the NF-κB pathway. Notably, the addition of exosomes in the presence of this NF-κB inhibitor further suppressed the expression of pp65, thus demonstrating the inhibitory effect of exosomes on the NF-κB pathway (Figure 5C–F). Furthermore, the combined exposure to this NF-κB inhibitor and exosomes had a greater effect on inhibiting the expression of pp65 than either treatment alone. Immunofluorescence staining confirmed the suppression of the NF-κB pathway by exosomes, which was also observed through a western-blotting analysis. By inhibiting the activation of the NF-κB pathway in chondrocytes, BMMSC-Exo and BMMSC-CA-Exo treatment subsequently reduced the production of inflammatory mediators and cytokines (Figure 5G). These findings suggest that exosomes can effectively inhibit the IL-1β-induced activation of the NF-κB signaling pathway in chondrocytes, resulting in the reduced production of inflammatory factors. Exposure to a combination of exosomes and a specific NF-κB pathway inhibitor further enhanced this anti-inflammatory effect. In addition, BMMSC-CA-Exo treatment was associated with a more significant decrease in the expression of inflammatory markers than BMMSC-Exo treatment.

### 2.6. Effect of BMMSC-Exo and BMMSC-CA-Exo Treatment on MAPK Pathways

This study used western-blotting techniques to assess the protein expression and phosphorylation levels of ERK, JNK, and p38 in chondrocytes treated with IL-1β to determine the influence of BMMSC-Exo treatment on the activity of proinflammatory signaling pathways. The results revealed that the phosphorylation levels of all three kinases were higher in the IL-1β-treated chondrocytes than in the controls (no IL-1β treatment). However, the phosphorylation levels of all three kinases were dramatically reduced in IL-1β-treated chondrocytes in the presence of BMMSC-Exo (Figure 6A,B). These findings suggest that BMMSC-Exo treatment has the potential to reduce the activity of proinflammatory signaling pathways in chondrocytes treated with IL-1β, which could have therapeutic potential for the treatment of inflammatory diseases such as OA.

### 2.7. Inhibition of c-Jun Phosphorylation via the JNK Pathway by BMMSC-Exo and BMMSC-CA-Exo Treatment of Chondrocytes

This study investigated the role of c-Jun phosphorylation in IL-1β-exposed chondrocytes. Since JNK is one of the major protein kinases responsible for the phosphorylation of c-Jun, we first examined its role in c-Jun phosphorylation in IL-1β-exposed chondrocytes. The results of the western blot analysis of JNK phosphorylation suggested that JNK activation and treatment with IL-1β resulted in the activation of JNK that appeared to occur before c-Jun expression and phosphorylation. However, BMMSC-Exo and BMMSC-CA-Exo treatment significantly decreased the expression of phosphorylated c-Jun (Figure 6A,B). These findings suggest that c-Jun expression is required for IL-1β to induce chondrocyte inflammatory expression and that c-Jun phosphorylation is necessary for this process. Furthermore, this study highlights the role of JNK in IL-1β-induced c-Jun phosphorylation and suggests that BMMSC-Exo and BMMSC-CA-Exo treatment can reduce the activity of the proinflammatory signaling pathways in chondrocytes that are treated with IL-1β. Overall, this study provides valuable insights into the mechanisms underlying the pathogenesis of OA and the potential therapeutic applications of exosomes in the treatment of this disease.

## 3. Discussion

The global prevalence of OA continues to escalate, both because of an aging population and the current obesity epidemic. OA is the most prevalent degenerative joint condition and is characterized by the gradual breakdown of articular cartilage and adult disability. It is often referred to as degenerative joint disease; however, this is a misnomer because OA is not a process of wear and tear but is a complex disease that involves multiple factors. OA is an age-related disorder that is more likely to develop as people age. Overweight and obesity are well-established risk factors for OA, which generally occurs in the knee but, also, to a lesser extent, in the hip and hand. Approximately 240 million individuals worldwide struggle with symptomatic OA, with 10% of men and 18% of women aged 60 years and older being affected [24]. Many challenges remain in the development of disease-modifying medicines for OA because the initiation and progression of OA include complicated molecular pathways of synovial inflammation, articular cartilage and subchondral bone degeneration, and bone remodeling. NSAIDs have been the primary treatment option for OA symptoms for a long time, and the mechanism by which NSAIDs exert their anti-inflammatory and analgesic effects is via the inhibition of the prostaglandin-generating enzyme, COX. In addition to their inflammatory potential, prostaglandins also contribute to important homeostatic functions, such as the maintenance of the gastric lining, renal blood flow, and platelet aggregation [25]. NSAIDs are unable to halt the progression OA, and extended use might induce adverse side effects. A recent study on PNs, a type of naturally occurring molecules, suggested their use as an alternative treatment for OA because of their anti-inflammatory properties and ability to stimulate ECM synthesis [26]. However, there remains an urgent need for therapies that can significantly improve the underlying etiology of OA while maintaining a high degree of safety. Natural compounds have demonstrated the ability to inhibit catabolic factors such as MMP and COX-2, while promoting anabolic factors such as COL-II and aggrecan. Some natural compounds can also modulate the cell cycle to protect against osteoarthritic cartilage degradation. Recent studies have shown that exosomes from BMMSCs reduce pain and inflammation in patients with OA [27,28]. Exosomes are small EVs that are secreted by most eukaryotic cells and play a vital role in intercellular communication by delivering a vast array of signaling molecules, such as mRNAs, miRNAs, nucleic acids, lipids, and proteins. Exosomes generated from numerous cell origins have been studied for their potential therapeutic use in a variety of diseases including OA.

Contemporary studies have assessed human bone-marrow-derived mesenchymal stromal cell (hBMSC)-derived EVs on IL-1β-stimulated chondrocytes obtained from patients with OA. One study aimed to evaluate the mechanism of action of the EVs, which have been proposed as a cell-free approach for the treatment of OA. EVs were found to attenuate the IL-1β-induced catabolic effects on chondrocytes by regulating the proinflammatory signaling pathways [29]. Furthermore, BMMSC-Exo offer a distinct therapeutic potential for the treatment of OA in that the bilipid layer of BMMSC-Exo preserves their endurance and stability, thereby maintaining the biological potency of the cell and protecting the cellular cargo from external degradation [30]. BMMSC-Exo have shown promising therapeutic potential for OA treatment by inhibiting the pyroptosis of cartilage by delivering miRNAs, particularly miR-326, which target HDAC3 and STAT1/NF-κB p65, to the chondrocytes [31]. In a study exploring the effects of exosomes isolated from human BMSCs on OA, differentially expressed lncRNAs and miRNAs were identified. A ceRNA network (LYRM4-AS1-GRPR-miR-6515-5p) was then selected for further investigation and was found to play a crucial role in regulating the protective effect of exosomes on OA via the LYRM4-AS1/GRPR/miR-6515-5p signaling pathway [32]. Together, these findings suggest that BMMSC-Exo have a unique therapeutic potential for OA treatment. BMMSC-Exo have been shown to promote cartilage repair, alleviate pain, inhibit the pyroptosis of cartilage, and regulate macrophage polarization [33]. Exosomes have shown efficacy in OA treatment by delivering therapeutic molecules associated with different signaling pathways that indirectly inhibit chondrocyte apoptosis. However, the isolation of exosomes is challenging because of their poor yield, which has hindered their clinical application. In addition, exosome drug loading has proven difficult as its effectiveness is limited by the low loading ability of exosomes [34]. Moreover, the isolation of exosomes is problematic because of their small and heterogeneous size, making their isolation and purification difficult. The presence of other EVs, such as microvesicles, can also complicate the isolation of exosomes. The improvement of exosome yield is a considerable hurdle for future applications, as the extraction of exosomes is difficult because of their differences in stability and easy destruction. However, researchers have made significant efforts to standardize the isolation, purification, and quantification of exosomes and to improve their targeting ability [35]. Several methods have been developed for exosome isolation including differential centrifugation, size-exclusion chromatography, ultrafiltration, and polymer precipitation [36]. Among these methods, a combination of different isolation methods may be better than the application of a single method. An ATPS-inclusive approach to exosome isolation involves the use of multiple techniques to improve the separation efficiency and enrichment of exosomes. ATPS isolation is a method that uses a two-phase system of polymers to separate the exosomes from other EVs based on their size and density. An ATPS is composed of two immiscible aqueous phases, typically PEG and DEX, which are combined to form two distinct phases. The selectivity of partitioning is often inadequate when the particles and contaminants have similar surface properties, which can complicate the isolation of exosomes using an ATPS. In one study, an ATPS-based isolation protocol was used to isolate small EVs from plant, cell culture, and mammalian cell sources [37]. These findings suggest that the use of an ATPS is a promising method for isolating exosomes with a high yield and purity. Exosome-depleted FBS was used before treatment with exosomes to avoid contamination with exogenous EVs that may have formed from the FBS in the culture medium. FBS is a blend of components that varies across batches and suppliers and may result in a substantial fluctuation in experimental outcomes. FBS has exogenous bovine EVs, RNA, and protein aggregates that may contaminate and alter the cargo composition of the experimental cell-derived EVs [38]. As this preliminary study focused on the effect of BMMSC-Exo in culture conditions, it should be emphasized that removing EVs from FBS drastically affected cell proliferation and differentiation. FBS may include unwanted materials such as endotoxins, mycoplasma, viral contamination, or prion proteins. Consequently, the exosome-depleted FBS is more effective than the standard FBS for exact exosome research and therapy. Finally, the exosome-depletion approach with 10% exosome-free FBS is performed optimally in the chondrocyte culture and later functional evaluations. This finding is in accordance with the culture techniques suggested by the International Society of Extracellular Vesicles (ISEV) [39].

In recent years, exosomes have gained considerable attention as potential therapeutic agents for various diseases including OA. However, the mechanism of action of exosomes has not been adequately discussed, and little is known about how exosomes exert their regenerative properties and efficient repair mechanisms. Our research has revealed that exosomes isolated from BMMSCs might have a major influence on the improved chondrocyte cell viability in response to IL-1β therapy, which suggests that BMMSC-Exo protect chondrocytes under certain conditions and in vitro pathogenic conditions of OA [40]. While conventional pharmacological interventions provide symptomatic relief, there is growing interest in exploring natural compounds with anti-inflammatory properties as potential therapeutic options. CA, a bioactive component extracted from cinnamon, has demonstrated its ability to inhibit key inflammatory pathways in OA. However, the therapeutic potential of exosomes derived from CA-treated cells has not been extensively investigated. Our research evaluated the mechanism of action of the effects of BMMSC-Exo and BMMSC-CA-Exo on hypertrophic chondrocytes (Figure 7). Furthermore, IL-1β can function as an independent proinflammatory cytokine in promoting the progression of OA by inducing chondrocyte apoptosis and proinflammatory signaling via the MAPK and NF-κB signaling pathways. The modulation of these proinflammatory signaling pathways was investigated to determine the molecular basis of these anti-inflammatory responses produced by BMMSC-Exo and BMMSC-CA-Exo in an IL-1β-stimulated in vitro OA model. The phosphorylation levels of ERK, JNK, p38, and NF-kB were assessed for this purpose. Our findings suggest that BMMSC-Exo can diminish the IL-1β-induced phosphorylation of ERK, JNK, p38, and NF-kB, thereby limiting the activity of pathways that involve these kinases. In addition, our preliminary data indicate that BMMSC-CA-Exo may exhibit enhanced anti-inflammatory effects compared with regular BMMSC-Exo, potentially due to the known anti-inflammatory properties of CA. However, further dedicated studies directly comparing BMMSC-Exo and BMMSC-CA-Exo are needed to confirm and quantify any additional anti-inflammatory advantages conferred by CA preconditioning. The inhibition of the MAPK and NF-κB pathways is a crucial exosome mechanism of action to regulate inflammation. The activation of macrophages by endothelial-cell-derived exosomes indicates the association of macrophage activation with the MAPK and NF-κB pathways [41]. The MAPK pathway involves the expression of various inflammatory mediators, while the NF-κB pathway plays a key role in the transcription of proinflammatory genes. By inhibiting these pathways, exosomes can reduce inflammation and alleviate various diseases including OA. These findings imply that the anti-inflammatory effects of BMMSC-Exo and BMMSC-CA-Exo may be induced via the regulation of IL-1β-induced proinflammatory signaling pathways.

### Future Challenges

While our study demonstrated the promising therapeutic potential of BMMSC-Exo and BMMSC-CA-Exo for OA treatment in an in vitro setting, several challenges must be addressed before clinical translation. First, comprehensive in vivo studies using relevant animal models of OA are crucial for evaluating the efficacy, optimal dosing, and delivery routes of these exosome preparations. Second, detailed mechanistic studies are warranted to elucidate the specific molecular pathways and cargo components (e.g., miRNAs and proteins) mediating the anti-inflammatory and chondroprotective effects. This could pave the way for engineering exosomes with an enhanced therapeutic potency. Furthermore, rigorous safety assessments, including the determination of potential off-target effects, immunogenicity, and long-term impacts, are imperative. Finally, successful clinical translation will require robust manufacturing protocols to ensure consistent exosome quality, potency, and scalability for human trials. Overcoming these challenges through systematic research will be critical for realizing the clinical potential of BMMSC-Exo with and without preconditioning with CA in OA management.

## 4. Materials and Methods

### 4.1. Antibodies and Reagents

Cell culture media 1× Dulbecco’s Modified Eagle’s Medium (DMEM), penicillin-streptomycin (pen-strep), and fetal bovine serum (FBS) were procured from Gibco (Thermo Fisher Scientific, Waltham, MA, USA). R&D Systems (Minneapolis, MN, USA) supplied human recombinant IL-1β, which was dissolved in PBS containing 0.5% bovine serum albumin (BSA). Antibodies against type II collagen (COL-II) (Abcam, Boston, MA, USA), COX-2 (Abcam, Boston, MA, USA), MMP-13 (Bioss, Woburn, MA, USA), p65 (cell signaling, Danvers, MA, USA), pp65 (cell signaling, Danvers, MA, USA), ERK (cell signaling, Danvers, MA, USA), pERK (cell signaling, Danvers, MA, USA), p38 (cell signaling, Danvers, MA, USA), pp38 (cell signaling, Danvers, MA, USA), JNK (cell signaling, Danvers, MA, USA), pJNK (cell signaling, Danvers, MA, USA), c-Jun (cell signaling, Danvers, MA, USA), pc-Jun (cell signaling, Danvers, MA, USA), CD63 (cell signaling, Danvers, MA, USA), CD81 (cell signaling, Danvers, MA, USA), and ALIX (Santa Cruz, TX, USA) were purchased. We also purchased goat anti-mouse IgG (ZyMax, Thermo Fisher Scientific, Waltham, MA, USA) and a conjugated goat anti-rabbit secondary antibody (H + L) (Novex life technologies, Thermo Fisher Scientific, Waltham, MA, USA). CA (≥95% purity, W228613) was purchased (Sigma-Aldrich, St. Louis, MO, USA) and dissolved in dimethyl sulfoxide (DMSO) to prepare a stock solution. (Refer Appendix A for details on CA concentration determination).

### 4.2. Source of Chondrocytes and BMMSC Culture

The human articular chondrocyte (HC-a) cell line and human BMMSC line were obtained from ScienCell (#4650, #7500), and the cells were cultured in 1× DMEM supplemented with 1% pen-strep and 10% heat-inactivated FBS at 37 °C in an atmosphere with 5% CO_2._ They were transferred to a culture plate and passed when the cell density reached 80%. Between passages 3 and 6, chondrocytes were used. Note: Please be aware that there are no donor criteria provided for HC-a. We suggest contacting the supplier, ScienCell, for further details on the characterization of HC-a cells [42,43]. However, passages 3–4 of BMMSCs were used in 1× DMEM supplemented with 1% pen-strep and 10% heat-inactivated exosome-free serum at 37 °C in an atmosphere with 5% CO_2_ for exosome isolation.

### 4.3. Exosome Isolation

BMMSCs were plated at a density of 5 × 10^5^ cells/well in 6-well cell culture plates with DMEM containing 10% FBS and 1% pen-strep at 37 °C in an atmosphere with 5% CO_2_. After reaching confluence, the BMMSCs were harvested 72 h after incubation with a serum-free exosome-containing medium with and without CA treatment. Exosomes in the cell supernatants were isolated by differential centrifugation. Centrifugation was performed at 5000× *g* for 5 min, followed by centrifugation at 7000–8000× *g* for 20 min at 4 °C. The pellet containing the exosomes was mixed with a PEG/DEX ATPS solution in a 1:1 volume ratio and incubated overnight at 4 °C. The pellet along with the ATPS was centrifuged at 5000× *g* for 5 min to separate the phases, and the DEX phase containing the exosomes was washed with ATPS wash solution until a sticky transparent layer was achieved. The purified exosomes were suspended in PBS and kept at −80 °C before lyophilization. The exosomes were lyophilized for use as treatments and stored at −80 °C.

### 4.4. Characterization Methods

#### 4.4.1. Characterization of the ATPS

To construct a binodal curve for the PEG/DEX system, the phase composition should be calculated, and the ATPS system should be characterized. The equilibrium phase was measured for volume and density by visually locating the phase separation lines and extracting the bottom phase with a micropipette. The interfacial layer of the ATPS was visualized using Coomassie Brilliant Blue R-250 to stain the PEG phase and the layers between phases.

#### 4.4.2. Nucleic Acid Analysis

Total RNA was extracted from the isolated exosomes using a conventional phenol-chloroform method. The exosome pellet was lysed using RNA isoplus (TaKaRa Bio Inc., Kusatsu, Japan), and the RNA was extracted according to the manufacturer’s protocol. The quantity and quality of the extracted RNA were assessed using Nanodrop.

#### 4.4.3. Protein Composition Analysis

The protein concentrations of the isolated exosome samples were measured using the bicinchoninic acid (BCA) method with a BCA Protein Assay kit. The kit standards and samples were loaded onto a 96-well plate and combined with the working reagent. The plate was then incubated for 30 min at 37 °C and analyzed with a spectrophotometer at 562 nm. The protein samples were preheated at 100 °C for 5 min; then, 8 µg of the sample was separated by polyacrylamide gel electrophoresis (SDS-PAGE) and transferred to a polyvinylidene difluoride (PVDF) membrane. After blocking with 5% nonfat milk for 90 min at 4 °C, the membranes were probed overnight with primary antibodies for CD81, CD63, and ALIX. After three washes in Tris-buffered saline with Tween20 (TBST) for 10 min, the membranes were incubated with horseradish peroxidase (HRP)-conjugated goat anti-mouse IgG (1:5000) for 2 h at 4 °C. They were washed again and incubated with chemiluminescent substrates. The blots were visualized using an enhanced chemiluminescence (ECL) system (Millipore, Bedford, MA, USA).

### 4.5. Evaluation of Cell Viability

A CCK-8 test kit (Dojindo, Kumamoto, Japan) was used to measure cell viability. (Refer Appendix A for details on CA cytotoxicity). The cells were seeded in 96-well plates at a density of 1 × 10^4^ cells/well for 24 h and then treated with BMMSC-Exo and BMMSC-CA-Exo at decreasing concentrations ranging from 500 to 0.03 µg/mL for 24 h. Following the proper treatment based on the experimental grouping, 10 µL of CCK-8 solution was added to each well and incubated for 2 h at room temperature (25 °C). Using a BioTek Synergy HTX multimode plate reader (Agilent Technologies, Santa Clara, CA, USA), the absorbance at 450 nm was detected for each well, and the amount of dye produced was determined. The CCK-8 test kit is a colorimetric assay kit that is used to measure cell proliferation and cytotoxicity. It is a ready-to-use solution that does not require the pre-mixing of components and offers a simple, rapid, reliable, and sensitive measurement of the cell viability and cytotoxicity of various chemicals. Although it measures the number of living cells, the CCK-8 assay has been used for cell viability and cytotoxicity assays. The detection sensitivity for CCK-8 is higher than that for other tetrazolium salts, such as MTT, XTT, MTS, and WST-1.

### 4.6. Role of Exosome in IL-1β-Induced Chondrocyte Inflammation

Human articular chondrocytes were seeded at a density of 5 × 10^5^ cells/well into 65-mm petri dishes in 1× DMEM containing 10% FBS and 1% pen-strep at 37 °C in an atmosphere with 5% CO_2_ and cultured for 24 h. Once the cells reached confluence, inflammation was induced by treatment with 10 ng/mL of IL-1β, and the exosome serum-free medium was changed. Lyophilized exosomes were reconstituted in PBS with a stock concentration of 10 mg/mL, from which 5 µg/mL of BMMSC-Exo and BMMSC-CA-Exo was used as a working concentration. Instead of pretreatment before stimulation with inflammatory cytokines, we chose to investigate post-treatment with exosomes following exposure to the proinflammatory cytokine IL-1β. This was decided because pre-treatment does not accurately reflect clinical outcomes. After incubation for 24 h, total RNA and total proteins were extracted from the chondrocytes using TRIzol reagent and radioimmunoprecipitation assay (RIPA) buffer.

### 4.7. PCR

Total RNA was extracted; then, the first-strand complementary DNA (cDNA) was synthesized using 0.5 μg of total RNA, and aliquots were reverse-transcribed in 20 μL of buffer containing 200 U of M-MLV reverse transcriptase, 0.25 mM DTT, and 250 μM each of dATP, dCTP, dGTP, and dTTP. Reverse transcription was performed using the following conditions: initial incubation at room temperature for 10 min, followed by 12 cycles of 25 °C for 30 s, 45 °C for 4 min, and 55 °C for 30 s, followed by 95 °C for 5 min and a 4 °C hold step in a GeneAmp PCR System 2700 (Applied Biosystems, Foster City, CA, USA). Next, aliquots of cDNA were amplified using AccuPower^®^ GreenStar PCR premix (Bioneer Co., Daejeon, Republic of Korea) in the MyGenie™ 96/384 thermal cycler system (Bioneer Co., Daejeon, Republic of Korea). Gene expression assays were performed for COL-II, COX-2, MMP-13, and GAPDH, and the expression levels for each gene were normalized and assessed compared with the expression of GAPDH (Refer Appendix A).

### 4.8. Whole-Cell Lysate Preparation and Western Blot Analysis

Once the treatment was complete, the cells were lysed with a RIPA lysis solution containing a protease and phosphatase inhibitor cocktail. The protein concentrations of the whole-cell lysates with and without exosome treatment were measured using a bicinchoninic acid (BCA) protein assay kit (Thermo Scientific, Waltham, MA, USA). The kit standards and samples were loaded onto a 96-well plate and combined with the working reagent. The plate was incubated for 30 min at 37 °C and analyzed with a spectrophotometer at 562 nm. The protein samples were incubated at 95 °C for 5 min, and 8 µg of each of the samples was separated by SDS-PAGE and transferred to a PVDF membrane. After blocking with 5% non-fat milk (DifcoTM, Becton Drive Franklin Lakes, NJ, USA) for 90 min at 4 °C, the membranes were probed overnight with primary antibodies for COX-2, COL-II, MMP-13, p65, pp65, ERK, pERK, p38, pp38, JNK, pJNK, c-Jun, pc-Jun, and the housekeeping gene GAPDH. After three 10 min washes with TBST, the membranes were incubated with the HRP-conjugated goat anti-mouse IgG (1:5000) and goat anti-rabbit IgG (H + L) highly cross-adsorbed secondary antibody for 2 h at 4 °C. The membranes were then washed with TBST three times at 10 min intervals. This washing step removes any unbound or nonspecifically bound proteins from the membrane. After washing, the membranes were incubated with chemiluminescent substrates. The incubation time for the substrate varies depending on the sensitivity of the substrate and the protein of interest. The incubation time typically ranges between 1–5 min. The chemiluminescent substrate reacts with the bound protein on the membrane and produces light, which can be detected using a luminometer. The resulting signal can be used to quantify the amount of protein present on the membrane. Blots were visualized using enhanced chemiluminescence (ECL) solution as per the manufacturer’s instructions. The band intensity was quantified using ImageJ software Version 1.54j (National Institutes of Health, Bethesda, MD, USA).

### 4.9. Scratch Wound Assay

Cell migration was evaluated using a scratch wound healing assay. Chondrocytes were seeded in 6-well plates and cultured to confluence. A scratch measuring approximately 1 mm in width was created in the confluent cell layer by using a 200 μL pipette tip. After two washes with PBS to remove loose cells, the control group was treated with medium alone, while the test group was treated with BMMSC-Exo and BMMSC-CA-Exo (5 μg/mL), and the plate was incubated at 37 °C. Images were obtained at 0 h, 24 h, and 48 h post-scratch.

### 4.10. Immunofluorescence

Chondrocytes were cultivated for 24 h on coverslips before the subsequent experiments were fixed with 4% paraformaldehyde in PBS for 15 min, permeabilized with 0.1% Triton X-100 for 15 min, and then blocked with 1% BSA in 0.1% Triton X-100 for 60 min. The coverslips were then incubated overnight at 4 °C with primary antibodies against COX-2 and COL-II at a 1:200 dilution and secondary antibodies at a 1:400 dilution. The cells were then washed with PBS, stained with phalloidin for 15 min, and washed again with PBS. They were mounted with DAPI and mounting medium, and fluorescence micrographs were obtained with a BioTek Lionheart FX Agilent microscope (Republic of Korea).

### 4.11. Inhibitor Treatment

Human chondrocytes (1 × 10^6^/mL) were allowed to grow for 2–3 days in 1× DMEM and FBS supplemented with penicillin (100 U/mL) before being used in the following experiments. Chondrocytes were serum-starved for 6 h to equilibrate all cells, and then pretreated with the NF-kB signaling inhibitor 10 μM 5HPP-33 (Calbiochem, La Jolla, CA, USA) for 2 h and exposed to 10 ng/mL of IL-1β in the presence or absence of 5 μg/mL of BMMSC-Exo and BMMSC-CA-Exo for 24 h.

### 4.12. Data Collection and Statistical Analysis

Statistical analysis was performed using GraphPad Prism software, version 9.0 (GraphPad Software, San Diego, CA, USA). One- or two-way analysis of variance was used for comparisons across several groups using parametric data. A *p*-value < 0.05 was statistically significant. All data are presented as the mean ± standard error of the mean.

## 5. Conclusions

According to our findings, BMMSC-Exo and BMMSC-CA-Exo inhibit the degeneration of COL-II, thereby preventing ECM degradation. The administration of BMMSC-Exo and BMMSC-CA-Exo significantly enhanced the transcriptional and translational expression of COL-II in IL-1β-exposed chondrocytes. As a defensive mechanism during inflammation, cytokines and inflammatory mediators are produced leading to the degradation of the ECM, and the ability of BMMSC-Exo and BMMSC-CA-Exo to alleviate inflammation may be associated with the decreased expression of inflammatory mediators such as COX-2, IL-6, and MMP-13. The expression of major inflammatory pathways, such as MAPK and NF-κB, when treated with BMMSC-Exo and BMMSC-CA-Exo, demonstrated an anti-inflammatory effect, suggesting that exosomes play a crucial role in regulating the expression of inflammatory markers. This can be used as a therapeutic approach to preventing the catabolism caused by inflammation and inhibiting the major inflammatory pathways involved in OA. Nevertheless, CA-treated exosomes could be used as a viable OA therapeutic approach, and further research is required to fully realize their molecular potential.

## Figures and Tables

**Figure 1 ijms-25-07263-f001:**
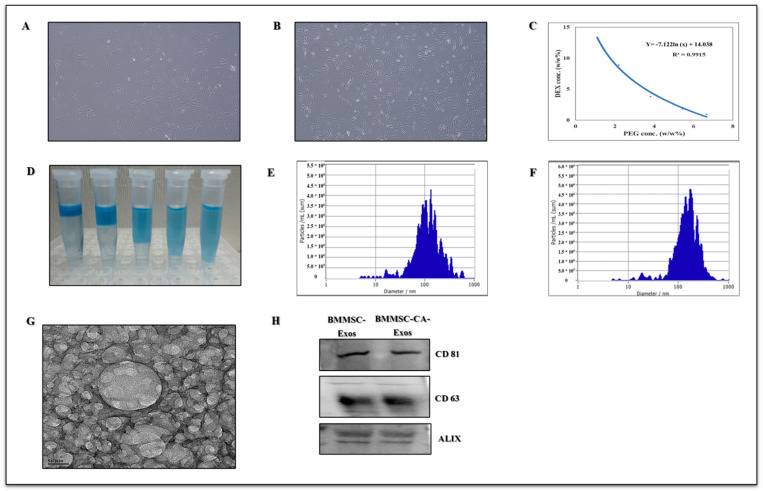
(**A**) Morphology of human BMMSCs. (**B**) Morphology of human chondrocytes (Magnification 4×). (**C**) Phase diagram of the PEG/DEX ATPS. The ATPS forms when the system concentration is above the binodal curve. (**D**) ATPS (to visualize the upper phase, Coomassie Brilliant Blue R-250 was added). (**E**) The size distribution of exosomes was identified by nanoparticle tracking analysis (NTA) (BMMSC-Exo). (**F**) The size distribution of exosomes was identified by NTA (BMMSC-CA-Exo). (**G**) TEM image of the exosomes (Scale bar 50 nm). (**H**) Western blot expression for the exosomal markers CD81, CD63, and ALIX.

**Figure 2 ijms-25-07263-f002:**
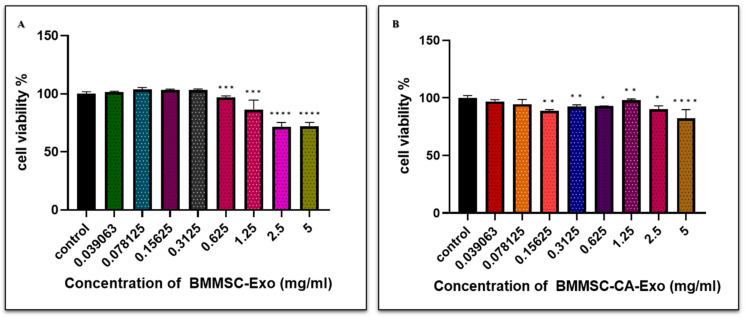
(**A**) Cell viability as a function of BMMSC-Exo and (**B**) BMMSC-CA-Exo treatment at decreasing concentrations ranging from 5 to 0.03 mg/mL. Data are presented as the mean ± standard deviation (*n* = 3). * *p* < 0.01, ** *p* < 0.05, *** *p* < 0.001, and **** *p* < 0.0001 compared with the control group.

**Figure 3 ijms-25-07263-f003:**
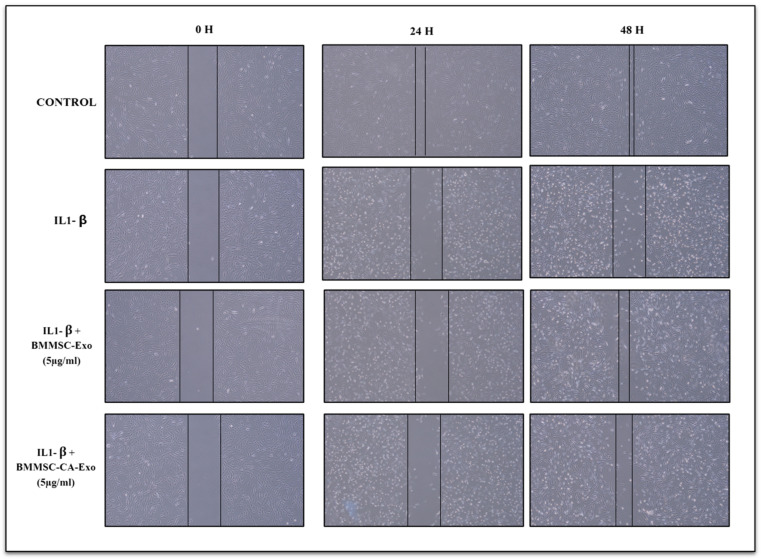
Migration rate of chondrocytes. Scratch wound assays reveal the migration rate of IL-1β-exposed chondrocytes stimulated with 5 μg/mL of BMMSC-Exo and BMMSC-CA-Exo (Scale bar 100 μm). Treatment with BMMSC-Exo and BMMSC-CA-Exo (5 μg/mL) significantly promoted the migration IL-1β-exposed chondrocytes compared with the induced inflammatory groups at 24 and 48 h.

**Figure 4 ijms-25-07263-f004:**
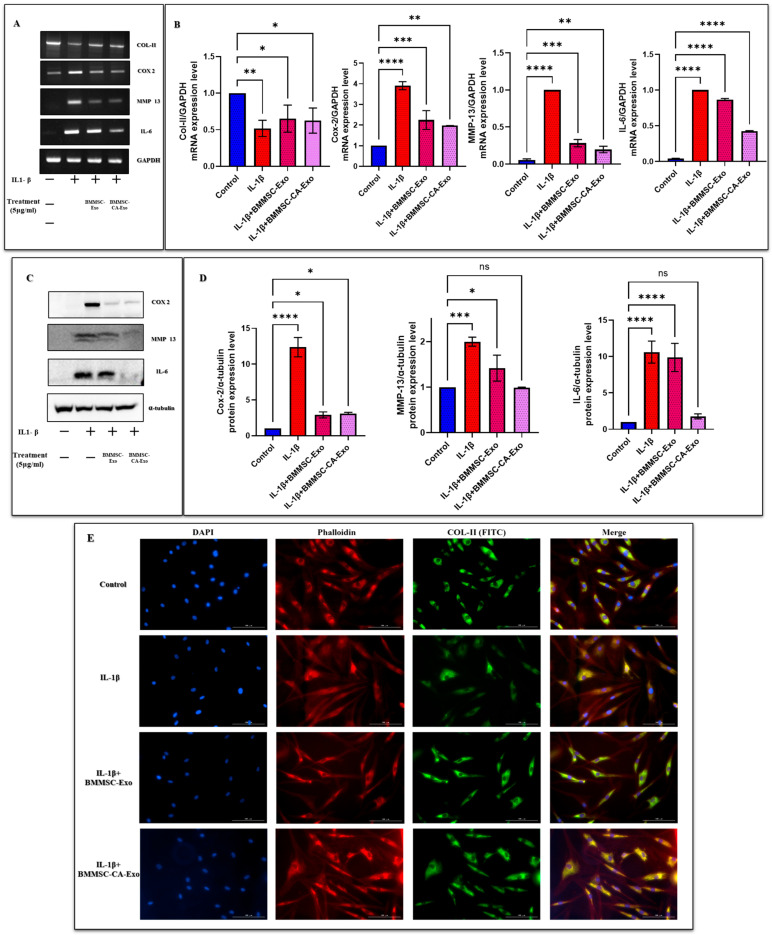
Effect of BMMSC-Exo and BMMSC-CA-Exo treatment on inflammation and cartilage-specific markers. (**A**) The gene expression levels of COL-II, COX-2, MMP-13, and IL-6 were analyzed using RT-PCR, (**B**) and quantitative analysis of the mRNA expression levels was performed for COL-II, COX-2, MMP-13, and IL-6. (**C**) The protein expression levels of COX-2, MMP-13, and IL-6 were determined by western blot analysis, and (**D**) a quantitative analysis was performed. (**E**) Immunofluorescence staining was performed to visualize the expression and localization of COL-II (Scale bar 100 µM). Data are presented as the mean ± standard deviation (*n* = 3) ns = non-significant, * *p* < 0.01, ** *p* < 0.05, *** *p* < 0.001, and **** *p* < 0.0001 compared with the nontreated group.

**Figure 5 ijms-25-07263-f005:**
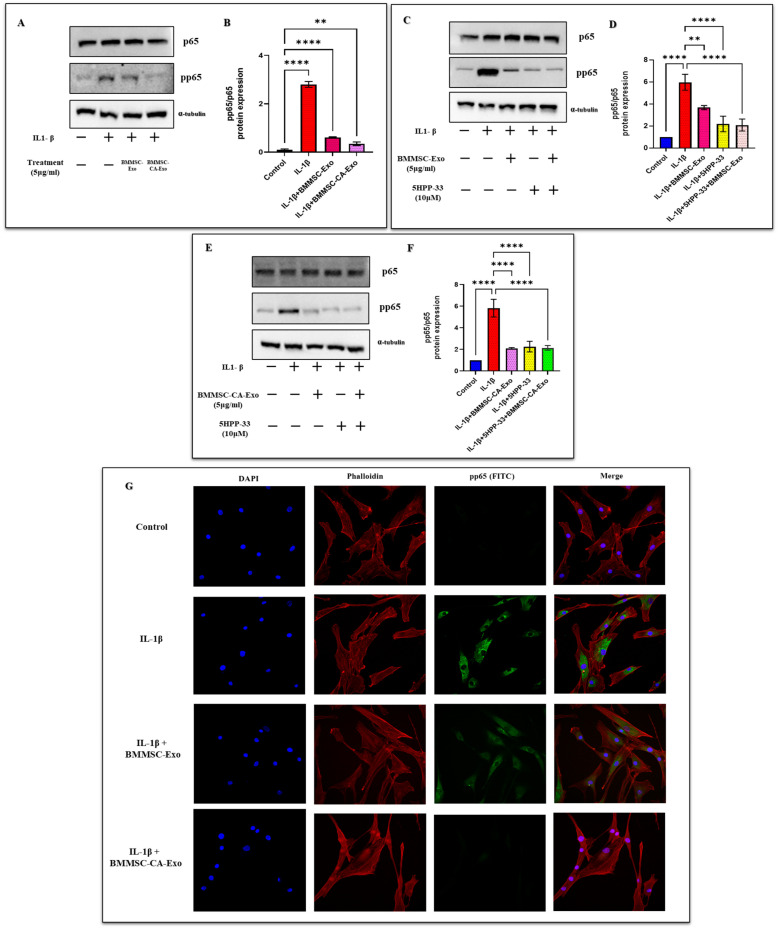
Effect of BMMSC-Exo treatment on NF-κB expression. (**A**) The protein expression levels of NF-κB (pp65) were determined by western blot analysis, (**B**) and a quantitative analysis was performed. (**C**–**F**) The protein expression levels of NF-κB (pp65) with the specific inhibitor 5HPP-33 were determined. (**G**) Immunofluorescence staining was performed to visualize the expression and localization of pp65 (Scale bar 100 µM). Data are presented as the mean ± standard deviation (*n* = 3). ** *p* < 0.05 and **** *p* < 0.0001 compared with the IL-1β-treated group.

**Figure 6 ijms-25-07263-f006:**
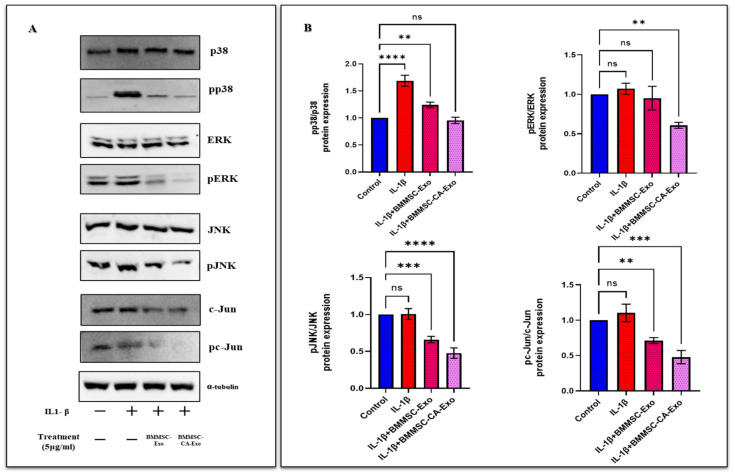
Effect of BMMSC-Exo and BMMSC-CA-Exo treatment on MAPK and c-Jun phosphorylation by the JNK pathway. (**A**) The protein expression levels of ERK, p38, JNK, and c-Jun were determined by western blot analysis. (**B**) A quantitative analysis of ERK, p38, JNK, and c-Jun was performed. Data are presented as the mean ± standard deviation (*n* = 3). ns = non-significant, ** *p* < 0.05, *** *p* < 0.001, and **** *p* < 0.0001 compared with the control group.

**Figure 7 ijms-25-07263-f007:**
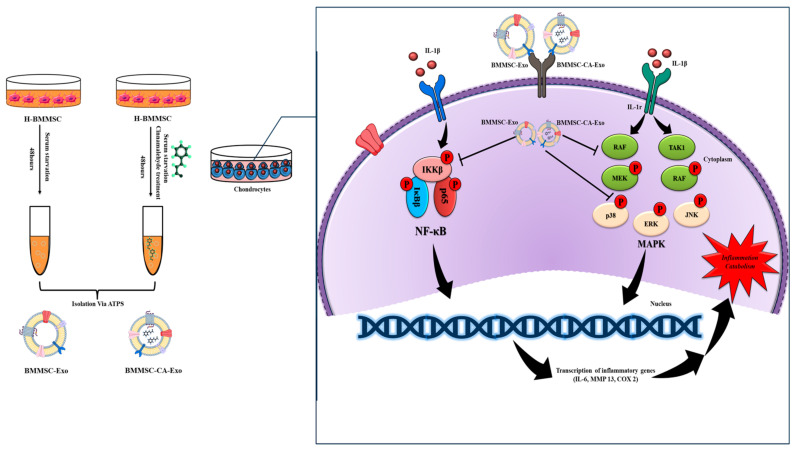
Schematic explanation of the mechanism by which BMMSC-Exo and BMMSC-CA-Exo inhibit the upregulation of p-NF-κB (pp65) expression in chondrocytes after IL-1β stimulation. IL-1β alters the transcriptional activity of MAPK by decreasing the expression of downstream mediators, such as IL-6, MMP-13, and COX-2. The effect of BMMSC-Exo and BMMSC-CA-Exo is exerted through the reduction of the expression of pNF-κB (p-p65) and pMAPK (pp38, pERK, and pJNK).

## Data Availability

All data generated or analyzed in this study are included in the article.

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
