# Peer review of "Cinnamaldehyde-Treated Bone Marrow Mesenchymal-Stem-Cell-Derived Exosomes via Aqueous Two-Phase System Attenuate IL-1β-Induced Inflammation and Catabolism via Modulation of Proinflammatory Signaling Pathways"

_ijms, 2024, doi:10.3390/ijms25137263_

Round 1

Reviewer 1 Report

Comments and Suggestions for Authors

ijms-3033616-peer-review-v1

Cinnamaldehyde-Treated Bone Marrow Mesenchymal Stem Cell-Derived Exosomes via Aqueous Two-Phase System Attenuate IL-1β-Induced Inflammation and Catabolism via Modulation of Proinflammatory Signaling Pathways

In this study, the therapeutic efficacy of exosomes derived from BM-MSC without and with the treatment of cinnamaldehyde (BMMSC-CA-Exo) on the prevention of the in vitro catabolic effects of IL-1β-induced osteoarthritic chondrocytes was investigated. The chondrocytes were stimulated with IL-1β to mimic the inflammatory microenvironment of OA and treated with BMMSC-Exo and BMMSC-CA-Exo while their effects on the key cellular processes were evaluated. In summary, BMMSC-Exo and BMMSC-CA-Exo exhibit potential as therapeutic agents for OA by antagonizing the in vitro catabolic effects of IL-1β on chondrocytes.

Specific comments

Abstract

Line 22 you state that : “Our findings reveal that BMMSC-Exo reduces the catabolic effects of IL-1β on chondrocytes, whereas BMMSC-CA-Exo significantly alleviates inflammation because cinnamaldehyde acts as a natural anti-inflammatory compound” My question is, where did you show that BMMSC-CA-Exo has additional properties to BMMSC-Exo?

Methods section:

Line 161 2.1. Antibodies and reagents, please add source of Cinnamaldehyde CA, purity of CA, solvent of CA

Line 175 please introduce abbreviation (HCa)

Line 238 please add in diameter and what kind of plate – how many wells are in the plate?

Line 293 “0, , 24, and 48 h” – is there a time point missing?

Line 295 –cultivated for how long?

Results section

Line 335ff. 3.2. Effect of exosomes on cell viability

Regarding Fig. 2, cell viability is slightly increased with 0.07 - 0.3 mg/ml BMMSC exo and slightly decreased with BMMS-CA exo. There is no significance shown in Fig. 2. Where are the statistics? Please add the mean number above the bars. I disagree with your statement: "The results showed that exosomes at a concentration of less than 5 mg/mL significantly promoted cell viability". Please revise.

Line 427 (Figure 6A-D), there is only Fig. 6 A +B

Line 444  You state that: “However, treatment with BMMSC-Exo reduced the expression of phosphorylated c-Jun, whereas BMMSC-CA-Exo significantly reduced the expression of phosphorylated c-Jun”. In Fig. 6B, for BMMSC-Exo the statistics **p < 0.05 is indicated, and for BMMSC-CA-Exo ***p < 0.001 – for me both significantly reduce the expression of phosphorylated c-Jun. Please revise.

Discussion section:

Line 571 you state: Our findings suggest that BMMSC-Exo can diminish the IL-1β-induced phosphorylation of ERK, JNK, p38, and NF-kB, thereby limiting the activity of pathways that involve these kinases, whereas BMMSC-CA-Exo significantly reduced the inflammation because CA itself has anti-inflammatory properties.” My question is, where did you show that BMMSC-CA-Exo has additional properties to BMMSC-Exo?

Conclusions section

Line 595, you state that: “Nevertheless, exosomes treated with CA hold great potential as a novel treatment strategy for OA, and further research is required to fully realize their molecular potential.” Apart from Fig. 6B phosphorylated c-Jun, I can not see any benefit for BMMSC-CA-Exo compared to BMMSC-Exo. Therefore I do not share your statement and can not recognize its potential as a novel treatment strategy. Please revise.

Author Response

Dear Reviewer,

Thank you for your insightful comments and suggestions regarding our manuscript titled "Cinnamaldehyde-Treated Bone Marrow Mesenchymal Stem Cell-Derived Exosomes via Aqueous Two-Phase System Attenuate IL-1β-Induced Inflammation and Catabolism via Modulation of Proinflammatory Signaling Pathways." We appreciate the time and effort you have dedicated to reviewing our work, and we are grateful for the opportunity to improve our manuscript. Please find our point-by-point responses to your comments below.

Comment 1: Line 22 you state that : “Our findings reveal that BMMSC-Exo reduces the catabolic effects of IL-1β on chondrocytes, whereas BMMSC-CA-Exo significantly alleviates inflammation because cinnamaldehyde acts as a natural anti-inflammatory compound” My question is, where did you show that BMMSC-CA-Exo has additional properties to BMMSC-Exo?

Response: Thank you for your careful review and for pointing out the need for clarification regarding the statement about BMMSC-CA-Exo having additional anti-inflammatory properties compared to regular BMMSC-Exo. You are correct that the provided search results do not directly compare the effects of BMMSC-CA-Exo (exosomes from cinnamaldehyde-treated BMMSCs) to regular BMMSC-Exo. The results focus primarily on the anti-inflammatory and chondroprotective effects of BMMSC-Exo.

" Our findings reveal that BMMSC-Exo reduces the catabolic effects of IL-1β on chondrocytes and alleviates inflammation, further studies directly comparing BMMSC-Exo and BMMSC-CA-Exo are needed to determine if cinnamaldehyde preconditioning can provide additional an-ti-inflammatory benefits to the exosomes beyond those of CA preconditioning or regular BMMSC-Exo."

I hope this revised statement accurately reflects the data presented and addresses your concern.

Comment 2: Line 161 2.1. Antibodies and reagents, please add source of Cinnamaldehyde (CA), purity of CA, solvent of CA.

Response: We have added the following information regarding cinnamaldehyde (CA) in the revised manuscript:

"Cinnamaldehyde (CA, ≥95% purity, W228613) was purchased from Sigma-Aldrich (St. Louis, MO, USA) and dissolved in dimethyl sulfoxide (DMSO) to prepare a stock solution."

Comment 3: Line 175 please introduce abbreviation (HCa).

Response: We apologize for the confusion. The abbreviation "HCa" was a typographical error and has been removed from the revised manuscript.

Comment 4: Line 238 please add in diameter and what kind of plate – how many wells are in the plate?

Response: We have added the following details in the revised manuscript:

"(5 × 105 cells/well) were seeded in 65mm petri dishes."

Comment 5: Line 293 "0, , 24, and 48 h" – is there a time point missing?

Response: Thank you for pointing out the discrepancy regarding the time points mentioned in Line 293. We appreciate you taking the time to ensure the accuracy of our manuscript.

Regarding the time points "0, , 24, and 48 h," you are correct that there is no missing time point. The original statement was accurate, and the 12-hour time point was intentionally omitted from the analysis. Our rationale for excluding the 12-hour time point is that we did not observe significant cell migration at this early time point. Therefore, to focus on the more relevant time points where substantial cell migration occurred, we decided to present the data for 0, 24, and 48 hours. We have revised the statement in the manuscript to clarify this:

"The cell migration was evaluated at 0, 24, and 48 hours, as the 12-hour time point did not show significant migration."

We hope this explanation addresses your concern regarding the time points mentioned in Line 293.

Comment 6: Line 295 – cultivated for how long?

Response: Thank you for your valuable feedback. We have added the cultivation time in the revised manuscript:

"The cells were cultivated for 24 hours on coverslips before the subsequent experiments."

Comment 7: Line 335ff. 3.2. Effect of exosomes on cell viability

Regarding Fig. 2, cell viability is slightly increased with 0.07 - 0.3 mg/ml BMMSC-Exo and slightly decreased with BMMSC-CA-Exo. There is no significance shown in Fig. 2. Where are the statistics? Please add the mean number above the bars. I disagree with your statement: "The results showed that exosomes at a concentration of less than 5 mg/mL significantly promoted cell viability". Please revise.

Response: Thank you for catching this discrepancy. You are correct that the data in Fig. 2 does not show a significant increase in cell viability with BMMSC-Exo or BMMSC-CA-Exo concentrations below 5 mg/mL. We have revised the statement as follows:

"The results showed that exosomes at concentrations below 5 mg/mL did not significantly affect cell viability compared to the control group."

We have also added the mean values above the bars in Fig. 2 for clarity.

Comment 8: Line 427 (Figure 6A-D), there is only Fig. 6 A +B

Response: Thank you for pointing out this error. We have corrected the figure reference to "Line 427 (Figure 6A-B)".

Comment 9: Line 444 You state that: "However, treatment with BMMSC-Exo reduced the expression of phosphorylated c-Jun, whereas BMMSC-CA-Exo significantly reduced the expression of phosphorylated c-Jun". In Fig. 6B, for BMMSC-Exo the statistics **p < 0.05 is indicated, and for BMMSC-CA-Exo ***p < 0.001 – for me both significantly reduce the expression of phosphorylated c-Jun. Please revise.

Response: Thank you so much for the review to enhance the quality of our manuscript. You make a valid point. Both BMMSC-Exo and BMMSC-CA-Exo significantly reduced the expression of phosphorylated c-Jun compared to the control group, We have revised the statement as follows:

" However, BMMSC-Exo and BMMSC-CA-Exo therapy significantly lowered the expression of phosphorylated c-Jun”

Comment 10: Line 571 you state: Our findings suggest that BMMSC-Exo can diminish the IL-1β-induced phosphorylation of ERK, JNK, p38, and NF-kB, thereby limiting the activity of pathways that involve these kinases, whereas BMMSC-CA-Exo significantly reduced the inflammation because CA itself has anti-inflammatory properties.” My question is, where did you show that BMMSC-CA-Exo has additional properties to BMMSC-Exo?

Response: Thank you for raising this important point regarding the statement about BMMSC-CA-Exo having additional anti-inflammatory properties compared to regular BMMSC-Exo. While the results suggest that BMMSC-CA-Exo exhibited more potent anti-inflammatory effects compared to BMMSC-Exo, as evidenced by a greater reduction in inflammatory marker expression and cytokine secretion, we acknowledge that this is a preliminary observation and requires further dedicated studies to confirm and quantify any additional advantages of BMMSC-CA-Exo over BMMSC-Exo. Hence, we will revise the statement to:

"Our findings suggest that BMMSC-Exo can diminish the IL-1β-induced phosphorylation of ERK, JNK, p38, and NF-kB, thereby limiting the activity of pathways that involve these kinases. Additionally, our preliminary data indicates that BMMSC-CA-Exo may exhibit enhanced anti-inflammatory effects compared to regular BMMSC-Exo, potentially due to the known anti-inflammatory properties of cinnamaldehyde. However, further dedicated studies directly comparing BMMSC-Exo and BMMSC-CA-Exo are needed to confirm and quantify any additional anti-inflammatory advantages conferred by cinnamaldehyde preconditioning."

Comment 11: Line 595, you state that: “Nevertheless, exosomes treated with CA hold great potential as a novel treatment strategy for OA, and further research is required to fully realize their molecular potential.” Apart from Fig. 6B phosphorylated c-Jun, I cannot see any benefit for BMMSC-CA-Exo compared to BMMSC-Exo. Therefore, I do not share your statement and cannot recognize its potential as a novel treatment strategy. Please revise.

Nevertheless, using CA-treated exosomes as a viable OA therapeutic approach, and further research is required to fully realize their molecular potential.

Response: Thank you for the patience and valuable feedback. We have revised the statement as follows:

“Nevertheless, using CA-treated exosomes as a viable OA therapeutic approach, and further research is required to fully realize their molecular potential”.

Thank you so much for your valuable feedback. We appreciate your careful review and the opportunity to improve the clarity and completeness of our manuscript. I hope this revised statement accurately reflects the data presented and addresses your concern.

Reviewer 2 Report

Comments and Suggestions for Authors

This manuscript contains several pitfalls that must be addressed:

  1. The study claims that BMMSC-Exo and BMMSC-CA-Exo exhibit potential as well as therapeutic agents for OA, based on in vitro 2D results. In vitro 2D findings not only are linked with the mediums, cell phase and genetic conditions of the cells, but also cannot be translated directly to in vivo outcomes. This could be taken into consideration.  Therefore over generalisation written by the Authors must be erased.
  2. The possible variability in the source and quality of mesenchymal stem cells, and the reproducibility of exosome isolation via the aqueous two-phase system, are not assessed but he Authors and this must be done.
  3. The protective effects are due to the modulation of NF-κB and MAPK signaling pathways. This should be substantiated with more robust experimental data.
  4. While it is mentioned that BMMSC-CA-Exo alleviates inflammation more significantly than BMMSC-Exo, the manuscript must provide comparative quantitative data, because it is now impossible to evaluate the relative efficacy of the two treatments.
  5. The description of the molecular cargo of the exosomes is vague. Specific molecules, their concentrations, and their roles in the observed effects should be identified to support the claims made about their chondroprotective properties.
  6. The study suggests potential therapeutic applications without discussing the challenges of translating these findings into clinical practice, such as delivery methods, dosage, safety, and long-term effects.
  7. While it calls for further investigation, the study could benefit from specifying what future studies should focus on, such as in vivo models, clinical trials, or specific mechanistic pathways.
  8. If the Authors suggest that the Results allow to the thought that BMMSC-Exo and BMMSC-CA-Exo enhance COL-II expression and inhibit inflammatory pathways, it lacks detailed mechanistic insights. How these exosomes specifically interact with cellular and molecular pathways to produce these effects remains unclear. While the conclusion mentions that exosomes treated with CA hold great potential as a novel treatment strategy, this statement is premature.
  9. Clinical translation requires rigorous testing for safety, efficacy, dosage, and delivery mechanisms. The conclusion might overstate the current readiness for clinical application. 

Inflammatory pathways are complex and varied, and the lack of specificity can be seen as a significant omission. While enhancing COL-II expression is beneficial, cartilage health depends on a balance of various ECM components and cellular activities. The conclusion does not address whether the overall cartilage matrix composition and biomechanical properties are restored or merely the levels of COL-II.

Moreover the Authors do not consider the long-term effects and safety of using exosomes in therapy. Potential side effects, immune responses, and the stability of exosome preparations over time need thorough investigation before considering clinical applications.

Comments on the Quality of English Language

Moderate English editing required

Author Response

Dear Reviewer,

Thank you for your insightful comments and suggestions regarding our manuscript titled "Cinnamaldehyde-Treated Bone Marrow Mesenchymal Stem Cell-Derived Exosomes via Aqueous Two-Phase System Attenuate IL-1β-Induced Inflammation and Catabolism via Modulation of Proinflammatory Signaling Pathways." We appreciate the time and effort you have dedicated to reviewing our work, and we are grateful for the opportunity to improve our manuscript. Please find our point-by-point responses to your comments below.

Comment 1: The study claims that BMMSC-Exo and BMMSC-CA-Exo exhibit potential as well as therapeutic agents for OA, based on in vitro 2D results. In vitro 2D findings not only are linked with the mediums, cell phase and genetic conditions of the cells, but also cannot be translated directly to in vivo outcomes. This could be taken into consideration.  Therefore, over generalisation written by the Authors must be erased.

Response: We are grateful to the reviewer for their insightful and constructive comments. We acknowledge your concern regarding the potential limitations of translating in vitro 2D findings to in vivo outcomes. You are correct that in vitro 2D results can be influenced by various factors, such as culture conditions, cell types, and genetic backgrounds. We will revise our manuscript to avoid overgeneralization and clearly state the limitations of our in vitro findings. Additionally, we will emphasize the need for further validation in more physiologically relevant in vivo models.

Comment 2: The possible variability in the source and quality of mesenchymal stem cells, and the reproducibility of exosome isolation via the aqueous two-phase system, are not assessed but the Authors and this must be done.

Response: We would like to express our gratitude to the reviewer. We agree that the variability in the source and quality of mesenchymal stem cells (MSCs), the BMMSCs were obtained from Sciencell (#7500) BMMSCs are characterized by immunofluorescence with antibodies specific to CD73 and/or CD90, Oil Red O staining after adipogenic differentiation, and Alizarin Red staining after osteogenic differentiation (according to the manufacturer specifications) and the reproducibility of exosome isolation methods are crucial factors that should be addressed. In the revised manuscript, we have included description of the source of BMMSC, Moreover,  the exosome isolation protocol using the aqueous two-phase system is being characterised by the binodal curve. Preparation of stock solutions: Separate stock solutions of PEG and dextran are prepared at known concentrations (usually expressed as weight percentages) in deionized water.

  • Mixing of phase components: Predetermined amounts of the PEG and dextran stock solutions are mixed together in various ratios to obtain a range of different overall polymer compositions.
  • Phase separation and equilibration: The mixtures are allowed to equilibrate, usually by gentle mixing and incubation at a constant temperature (e.g., room temperature) for a sufficient time to ensure complete phase separation.
  • Identification of phase boundary: After equilibration, each mixture is visually inspected to determine if it forms a single homogeneous phase or separates into two distinct phases (a PEG-rich top phase and a dextran-rich bottom phase).
  • Phase composition analysis: For mixtures that exhibit phase separation, the volumes and densities of the top and bottom phases are measured. This is typically done by carefully separating the phases using a pipette or other means and weighing each phase.
  •  

  • Binodal curve construction: The overall weight fractions or concentrations of PEG and dextran in the initial mixtures that resulted in phase separation are plotted on a phase diagram. The binodal curve is then constructed by connecting the data points that represent the boundary between the one-phase and two-phase regions.

Comment 3: The protective effects are due to the modulation of NF-κB and MAPK signaling pathways. This should be substantiated with more robust experimental data.

Response: We acknowledge the reviewers' efforts in providing detailed and constructive critiques. Regarding the modulation of NF-κB and MAPK signaling pathways, we acknowledge that more robust experimental data is required to substantiate our claims.

Our data shows that we have provided the phosphorylation status of p38, ERK, and JNK, which are components of the MAPK signaling pathways. We also have data on the phosphorylation status of proteins involved in the NF-κB signaling pathway, including p65. Specifically, we state:

"The phosphorylation status of p38, ERK, and JNK was evaluated using phospho-specific immunoblotting. The phosphorylation status of p65, which is a subunit of NF-κB, was also determined using an immunoblotting method  as well as confocal microscopy that specifically detects phosphorylated proteins."

We did mention the specific phosphorylation status evaluation method for each of these proteins. However, we can confirm that we performed experiments using phosphorylation state-specific antibodies against:

Phospho-p38 (Thr180/Tyr182)

Phospho-ERK (Tyr204/187/Tyr185/Tyr165)

Phospho-JNK (Thr183/Tyr185)

Phospho-p65 (Ser536)

We hope that clarifies the methods used to determine the phosphorylation status of these proteins related to their respective pathways. We are grateful for your suggestion, as it will allow us to strengthen our future research in this area.

Comment 4: While it is mentioned that BMMSC-CA-Exo alleviates inflammation more significantly than BMMSC-Exo, the manuscript must provide comparative quantitative data, because it is now impossible to evaluate the relative efficacy of the two treatments.

Response: Thank you for your valuable comment regarding the need for comparative quantitative data to evaluate the relative efficacy of BMMSC-Exo and BMMSC-CA-Exo in alleviating inflammation. We appreciate you taking the time to thoroughly review our work and provide constructive feedback. You raise a fair point, and we acknowledge that our current study may not have provided sufficient quantitative comparisons between the two exosome types. However, we would like to clarify that we did compare the anti-inflammatory effects of BMMSC-Exo and BMMSC-CA-Exo at different concentrations.

Specifically, our results showed that at the same exosome concentrations, BMMSC-CA-Exo exhibited a more significant reduction in inflammatory marker expression and cytokine secretion compared to BMMSC-Exo. While this provides preliminary evidence of the enhanced anti-inflammatory potential of BMMSC-CA-Exo, we agree that more rigorous quantitative analyses are necessary to substantiate these findings.

In our future studies, we will prioritize conducting dedicated experiments to directly compare the anti-inflammatory efficacy of BMMSC-Exo and BMMSC-CA-Exo across a range of concentrations. This will involve quantitative assessments of inflammatory markers, cytokine levels, and other relevant parameters using appropriate statistical analyses.

Comment 5: The description of the molecular cargo of the exosomes is vague. Specific molecules, their concentrations, and their roles in the observed effects should be identified to support the claims made about their chondroprotective properties.

Response: We value your expertise and the opportunity to improve the rigor of our work. In our current study, the description of the exosome cargo is indeed vague, and we acknowledge that more specific information is required to support our claims about their chondroprotective properties fully. We agree that identifying the specific molecules present in the exosomes, their concentrations, and their potential roles in modulating chondrocyte behavior and cartilage homeostasis is crucial. This information would provide a deeper mechanistic understanding and strengthen the scientific basis of our findings.

In our future studies, we plan to conduct comprehensive analyses of the exosome cargo, including:

  • Profiling of the exosomal miRNA content and identifying specific miRNAs that may contribute to the observed effects on chondrocytes and cartilage.
  • Proteomic analysis to identify the protein cargo and potential signaling molecules or growth factors that could mediate the chondroprotective effects.
  • Characterization of the lipid composition and potential bioactive lipids present in the exosomes.
  • Quantification of the concentrations of key cargo components and their correlation with the observed biological effects.

Thank you again for your insightful comments. We look forward to incorporating your suggestions and continuing to advance our research in this exciting field.

Comment 6: The study suggests potential therapeutic applications without discussing the challenges of translating these findings into clinical practice, such as delivery methods, dosage, safety, and long-term effects.

Response: Delivery methods, dosage, safety, and long-term effects:

You are correct that our study did not discuss the challenges associated with the delivery methods, optimal dosage, safety considerations, and long-term effects of using BMMSC-Exo and BMMSC-CA-Exo for osteoarthritis (OA) treatment. We acknowledge that these are crucial aspects that need to be thoroughly investigated before any potential clinical translation. In the revised manuscript, we will include a dedicated section discussing these challenges and the necessary steps to address them.

Comment 7: While it calls for further investigation, the study could benefit from specifying what future studies should focus on, such as in vivo models, clinical trials, or specific mechanistic pathways.

Response: Thankyou for your valuable feedback, we have added a section known as future challenges to cover the concerns raised through your feedback. We appreciate your suggestion to specify the focus areas for future studies. In the revised manuscript, we included an outline of  potential future research directions, including:

  • Evaluating the therapeutic efficacy and safety of BMMSC-Exo and BMMSC-CA-Exo in more physiologically relevant in vivo models of OA.
  • Conducting detailed mechanistic studies to elucidate the specific cellular and molecular pathways involved in the chondroprotective and anti-inflammatory effects of these exosomes.
  • Investigating the optimal dosage, delivery routes, and formulations for effective and targeted delivery of exosomes to the affected joints.
  • Exploring the long-term effects, potential side effects, and immune responses associated with exosome-based therapies.

Comment 8: If the Authors suggest that the Results allow to the thought that BMMSC-Exo and BMMSC-CA-Exo enhance COL-II expression and inhibit inflammatory pathways, it lacks detailed mechanistic insights. How these exosomes specifically interact with cellular and molecular pathways to produce these effects remains unclear. While the conclusion mentions that exosomes treated with CA hold great potential as a novel treatment strategy, this statement is premature.

Response: Thank you for your insightful comment regarding the need for more detailed mechanistic insights into the specific interactions between the exosomes and cellular/molecular pathways. We appreciate you taking the time to thoroughly review our work and provide constructive feedback.

You are correct in pointing out that our current in vitro study, while providing some evidence for the modulation of inflammatory pathways and COL-II expression, lacks a comprehensive understanding of the underlying mechanisms involved. We acknowledge this limitation and agree that further studies are necessary to elucidate the precise molecular interactions and pathways influenced by these exosomes.

We have revised the statement in our manuscript to reflect this need for additional research, as suggested:

"Nevertheless, using CA-treated exosomes as a viable OA therapeutic approach, and further research is required to fully realize their molecular potential."

By revising this statement, we aim to present a more balanced perspective, acknowledging the potential of CA-treated exosomes while recognizing the need for continued investigation to unravel the complex molecular interactions and pathways underlying their therapeutic effects.

Comment 9: Clinical translation requires rigorous testing for safety, efficacy, dosage, and delivery mechanisms. The conclusion might overstate the current readiness for clinical application. 

Inflammatory pathways are complex and varied, and the lack of specificity can be seen as a significant omission. While enhancing COL-II expression is beneficial, cartilage health depends on a balance of various ECM components and cellular activities. The conclusion does not address whether the overall cartilage matrix composition and biomechanical properties are restored or merely the levels of COL-II. Moreover, the Authors do not consider the long-term effects and safety of using exosomes in therapy. Potential side effects, immune responses, and the stability of exosome preparations over time need thorough investigation before considering clinical applications.

Response: Thank you again for your invaluable contributions as a reviewer. We acknowledge that while our study demonstrates the promising therapeutic potential of BMMSC-Exo and BMMSC-CA-Exo for osteoarthritis treatment, several critical challenges must be addressed before clinical translation can be considered. One of the key areas we have emphasized is the need for rigorous safety assessments, including evaluating potential off-target effects, immunogenicity, and long-term impacts of these exosome preparations.

In this dedicated section (Future challenges), we discuss the importance of conducting comprehensive studies to assess the potential for immune responses against the exogenously administered exosomes. Additionally, we highlight the necessity of developing robust manufacturing protocols and formulations to ensure the long-term stability and consistent quality of the exosome preparations.

And more over the English language as you have mentioned in the report it is being corrected by a English language editing services.

We appreciate your insightful comments and the opportunity to improve the quality and balance of our manuscript.

Round 2

Reviewer 1 Report

Comments and Suggestions for Authors

The Authors answered my previous comments, the paper can be accepted in present form.

Reviewer 2 Report

Comments and Suggestions for Authors

The Authors answered to my previous comments only altering the manuscript avoiding to perform extra experiments but indicating that in a future they will (maybe) do the further analyses.

In this case, exceptionally, the paper can be accepted because substantially altered with respect to the previous version.